# Antiphospholipid Antibodies Occurrence in Acute SARS-CoV-2 Infection without Overt Thrombosis

**DOI:** 10.3390/biomedicines11051241

**Published:** 2023-04-22

**Authors:** Alina Dima, Daniela Nicoleta Popescu, Ruxandra Moroti, Elisabeta Stoica, Georgiana State, Florentina Negoi, Ioana Adriana Berza, Magda Parvu

**Affiliations:** 1Department of Rheumatology, Colentina Clinical Hospital, 020125 Bucharest, Romania; 2Department of Infectious Diseases, Carol Davila University of Medicine and Pharmacy, 020021 Bucharest, Romania; 3Department of Infectious Diseases, National Institute for Infectious Diseases Matei Bals, 021105 Bucharest, Romania

**Keywords:** antiphospholipid syndrome, criteria antiphospholipid antibodies, lupus anticoagulant, C-reactive protein, COVID-19 severity, SARS-CoV-2

## Abstract

We sought to determine the prevalence of antiphospholipid antibodies (aPLs) and their correlation with COVID-19 severity (in terms of clinical and laboratory parameters) in patients without thrombotic events during the early phase of infection. This was a cross-sectional study with the inclusion of hospitalized COVID-19 patients from a single department during the COVID-19 pandemic (April 2020–May 2021). Previous known immune disease or thrombophilia along with long-term anticoagulation and patients with overt arterial or venous thrombosis during SARS-CoV-2 infection were excluded. In all cases, data on four criteria for aPL were collected, namely lupus anticoagulant (LA), IgM and IgG anticardiolipin antibodies (aCL), as well as IgG anti-β2 glycoprotein I antibodies (aβ2GPI). One hundred and seventy-nine COVID-19 patients were included, with a mean age of 59.6 (14.5) years and a sex ratio of 0.8 male: female. LA was positive in 41.9%, while it was strongly positive in 4.5%; aCL IgM was found in 9.5%, aCL IgG in 4.5%, and aβ2GPI IgG in 1.7% of the sera tested. Clinical correlation: LA was more frequently expressed in severe COVID-19 cases than in moderate or mild cases (*p* = 0.027). Laboratory correlation: In univariate analysis, LA levels were correlated with D-dimer (*p* = 0.016), aPTT (*p* = 0.001), ferritin (*p* = 0.012), C-reactive protein (CRP) (*p* = 0.027), lymphocyte (*p* = 0.040), and platelet (*p* < 0.001) counts. However, in the multivariate analysis, only the CRP levels correlated with LA positivity: OR (95% CI) 1.008 (1.001–1.016), *p* = 0.042. LA was the most common aPL identified in the acute phase of COVID-19 and was correlated with infection severity in patients without overt thrombosis.

## 1. Background

While most patients diagnosed with the new severe acute respiratory syndrome coronavirus 2 (SARS-CoV-2) disease (COVID-19) are asymptomatic or have mild disease, approximately 14% of cases are severe, developing acute respiratory failure, and 5% to 8% need admission to intensive care units [1].

The poor prognosis of COVID-19 is related not only to pulmonary direct viral invasion but also to the associated systemic (hyper)-inflammatory immune response, including coagulation disturbances. Therefore, increased D-dimer levels were reported to be correlated with COVID-19 severity [2], and the benefits of heparin or other anticoagulants in lowering mortality were noted [3]. The coagulation abnormalities correspond to micro- and macrothrombosis, including pulmonary thromboembolism (PTE) in severe cases [4].

The occurrence of immune phenomena, antiphospholipid antibodies (aPLs) included, has been described in many viral infections, such as those with human immunodeficiency virus (HIV), Epstein-Barr virus, cytomegalovirus, parvovirus, B or C hepatitis viruses (HBV, HCV) [5], and recently SARS-CoV-2 [6,7]. The significance of aPL positivity in viral infections has not been fully characterized, as their positivity is often transitory and without overt clinical expression [5,8].

Antiphospholipid syndrome (APS) is defined according to the Sydney 2006 criteria by the presence and persistence of at least 12 weeks of criteria aPL, including lupus anticoagulant (LA), anticardiolipin antibodies (aCL), or anti-β2 glycoprotein I antibodies (aβ2GPI) in patients with a history or ongoing thrombotic events (arterial/venous) or pregnancy pathology [9].

The COVID-19-related mechanism of associated thrombophilia is complex and not fully understood, and the involvement of aPL in the thrombotic phenomena related to SARS-CoV-2 infection remains a matter of debate [6,7]. Even in the case of aPL positivity, LA as well as “solid-phase” antibodies, such as aCL and aβ2GPI, might be a feature of the SARS-CoV-2 infection as previously reviewed; the data regarding their role in thrombosis occurrence during COVID-19 remains scarce and inconclusive [6,7].

Therefore, the present study aimed to evaluate the prevalence and clinical correlations of criteria aPLs in a subset of COVID-19 patients without thrombotic events at admission.

## 2. Methods

### 2.1. Study Population

We herein present a cross-sectional study that included adult patients (more than 18 years) with COVID-19 hospitalized in a dedicated COVID-19 department between April 2020 and May 2021.

The diagnosis of COVID-19 was confirmed in all cases by real-time (RT) polymerase chain reaction (PCR) SARS-CoV-2 using specimens derived from nasopharyngeal swabs.

Furthermore, patients with a history of thrombophilia or autoimmune diseases or chronic anticoagulant treatment and patients with intra-COVID-19 thrombotic events were excluded from the analysis. In most patients, preventive anticoagulant treatment was added during follow-up, but treatment was started only after the first blood sample at admission (APS serology included).

Clinical data: The medical records included the COVID-19 onset data, signs and symptoms, treatment, and outcome. After the introduction of the COVID-19 vaccination, the patients’ vaccination status was recorded.

Biological data: The laboratory tests at admission were standardized in our department; therefore, they were uniformly performed initially in all patients and further extended in selected cases depending on case particularity (e.g., troponin, NT pro-BNP). APS serology (our hospital laboratory had available determination for LA, aCL IgM, aCL IgG, and aβ2GPI IgG) was one of the blood tests included in the standard care of COVID-19 patients.

### 2.2. Variables

Clinical and laboratory data were collected from the patient’s hospital files. For all patients, the same clinical and biological data were registered to avoid altering the group homogeneity.

The aPLs were detected by an enzyme-linked immunosorbent assay (ELISA) (QUANTA Lite^®^ ELISAs provided by INOVA Diagnostics, San Diego, CA, USA) with internal standard HCAL for IgG and EY2C9 for IgM. The ELISA kit used purified CL or β2GPI antigen immobilized in microplate wells. IgG and IgM classes of aCL and aβ2GPI in patient sera were further bound to the immobilized antigen and added enzyme-labeled anti-human IgG or IgM conjugate, which bind to the patient antibodies when present. The laboratory cut-off values for positivity were set as 20 microgram of IgM anticardiolipin antibody per liter (MPL) for aCL IgM, 20 microgram of IgG anticardiolipin antibody per liter (GPL) for aCL IgG, and 20 microgram of IgG anti-beta2 glycoprotein I antibody per liter (GBU) for aβ2GPI IgG.

LA was detected by the standardized procedure based on the guidelines of the International Society on Thrombosis and Hemostasis [10] using the STA R Max^®^ hemostasis analyzer developed by Diagnostica Stago S.A.S (Stago). The LA-integrated tests included screening and confirmation in a single procedure, using two reagents STA clot DRVV screen and STA clot DRVV confirm. The results were calculated by the LA ratio (screen/confirm). The laboratory cut-offs used to define LA positivity were as follows: LA < 1.2 negative, LA 1.2–1.5 weakly positive, LA 1.5–2.0 positive, and LA > 2.0 strongly positive.

The severity of COVID-19 pulmonary involvement was classified as absent, mild (less than 25%), moderate (25–50%), severe (50–75%), and very severe (>75%), depending on the chest computed tomography (CT)scan results [11]. In all patients with clinical and/or laboratory findings that raised a suspicion of PTE, a contrast-enhanced exam was added to the native CT scan, and patients who developed thrombosis were excluded from our analysis.

Severe COVID-19 forms were defined by breath shortening with polypnea (>30 breaths/minute), low oxygen saturation (<93%), more than 50% lung involvement on a pulmonary CT scan, or multiple organ dysfunction [12].

The Hospital’s Ethics Committee approved the research presented here (no 3/2022).

### 2.3. Statistical Analysis

Categorical values are expressed according to their distribution as numbers and percentages, while quantitative values are expressed as the mean ± standard deviation (SD) or median (med) with the interquartile range (IQR) q25 first quartile, q75 third quartile, and nominal variables as percentages. According to the variable’s type, a bivariate analysis using the Mann–Whitney test (and Kruskal-Wallis test with more than two groups) or Chi-square test was performed. The univariate analysis was tested by Spearman’s rho. Parameters associated with a *p*-value less than 0.05 in bivariate tests were then tested in multivariate analysis by binary logistics. IBM SPSS Statistics V25 software was used.

## 3. Results

### 3.1. Descriptive

Data from 179 consecutive COVID-19 patients fulfilling the inclusion and exclusion criteria were collected. The mean (SD) age at inclusion was 59.6 (14.5) years old, with 99 (55.3%) females. Seventy-two (40.2%) patients had severe COVID-19 (Table 1).

None of the patients included had been vaccinated against COVID-19. Cough, fever, and myalgia were the most frequent COVID-19 symptoms, reported in 114 (63.7%), 106 (59.2%), and 75 (41.9%) patients, respectively (Appendix A). Additionally, at least two comorbidities were associated in 88 (49.2%) patients (Appendix A).

Laboratory data regarding parameters such as hemogram, C-reactive protein (CRP), ferritin, interleukin-6 (IL-6), troponin, NT-proBNP, albumin, and 25OH-vitamine D are also presented (Appendix A).

### 3.2. Antiphospholipid Syndrome Serology

The median (q1; q3) time from COVID-19 symptom onset and SARS-CoV-2 RT-PCR diagnosis was 8 (5; 10) days and 3 (1; 6) days, respectively (Table 1).

LA positivity was the most frequent aPL identified, was moderately positive in 74 (41.9%) patients, and was strongly positive in 8 (4.5%) patients, followed by aCL IgM positivity in 17 (9.5%) patients (Appendix A).

The aCL IgM titers were correlated with the aCL IgG and aβ2GPI IgG titers, *p* < 0.001, rho 0.408 and *p* < 0.001, rho 0.282, respectively (Appendix A). The levels of aCL IgM and aβ2GPI IgG were significantly higher in patients with positive LA (Table 2).

Regarding the relation with the coagulation parameters, LA levels correlated weakly with the D-dimer (*p* = 0.016; rho = 0.183) and aPTT values (*p* = 0.001; rho = 0.257) and inversely with platelet count (*p* < 0.001; rho = −0.265) (Appendix A).

It is noted that in patients with positive aCL IgM, the levels of IL-6 were significantly higher: 45.7 (36.7; 226.9) ng/mL versus 24.2 (9.8; 55.5) ng/mL, *p* = 0.029 (Appendix A).

When analyzing disease severity, LA levels were significantly different between patients with mild versus moderate or severe disease, but not those of aCL IgM, aCL IgG, or aβ2GPI IgG (Appendix A).

Additionally, levels of the laboratory parameters related to COVID-19 disease severity were significantly higher (ferritin, CRP) or lower (lymphocytes) in patients with positive LA: 504 (240; 815) ng/mL versus 143 (89; 436) ng/mL; *p* = 0.012, 44.2 (10.8; 85.8) mg/dL versus 13.1 (5.7; 19.9) mg/dL; *p* = 0.027, and 960 (710; 1380)/µL versus 1870 (940; 2410)/µL; *p* = 0.040, respectively (Appendix A).

In multivariate analysis, the CRP levels predicted LA positivity: OR (95% CI) 1.008 (1.001–1.016), *p* = 0.042. (Appendix A).

## 4. Discussion

The present study searched for the prevalence and clinical correlation of diagnostic aPL in patients with ongoing COVID-19 who did not have overt arterial or venous thrombosis at admission for acute SARS-CoV-2 infection.

APS is an autoimmune thrombophilia defined by the development of circulating aPL with pathogenic pro-thrombotic properties, and both persistent serum criteria aPL and arterial or venous thrombosis are necessary conditions for sustaining the diagnosis [13].

In COVID-19, venous and arterial thrombosis occurrence could be viewed through the prism of Virchow’s triad with further endothelial dysfunction, platelet activation, hyperviscosity, and blood flow abnormalities due to hypoxia, immune reactions, and hypercoagulability [3].

During screening, aPLs are also found in healthy subjects. However, when compared to controls, the rates of aPL positivity are significantly higher in COVID-19 cases, such as 27.6% versus 2.5% for IgM aCL in one study [14]. Of note, as in other immune or viral pathologies, the reported prevalence for aPLs varies largely in COVID-19. One reason might be the methods used for analysis, e.g., aPL is more likely detectable, especially in cases with low titers, by ELISA than by multiplexed fluorometric immunoassay (MFIA) [14], which underlines the need to standardize the laboratory methods used for aPL [15]. The different time intervals elapsed from the onset of infection to the search for aPL could also influence the results obtained. We herein present data from patients tested approximately eight days after the onset of SARS-CoV-2 symptoms.

SARS-CoV-2 interacts with both innate and acquired immunity and determines the production of various autoantibodies [16]. In COVID-19, the reported prevalence for LA, aβ2GPI IgM, aβ2GPI IgG, aCL IgM, and aCL IgG was 33%, 4–10%, 12%, 4%, and 11%, respectively [15,17]. When compared to other viral or bacterial infections, in COVID-19, a 53% overall prevalence of positive aPLs was reported versus 49% in other viral/bacterial infections [15]. Moreover, antinuclear antibodies have also been reported in approximately 25% of COVID-19 cases [18].

Similar to SARS-CoV-2, aCL appearance during uncomplicated viral or bacterial infections is already known. High rates of positivity were particularly reported in HIV, HBV, and HVC (50%, 25%, and 20%, respectively) [5]. Additionally, a seven-fold increase for HIV and a three-fold increase for HBV or HCV in the risk for thromboembolic events was reported in these patients. Further, regarding bacterial infections, the highest frequency of aPL positivity was described during Coxiella burnetii, Mycoplasma pneumonie, and Streptococci infections, with an occurrence of aPL in 17.2%, 2.2%, and 2.2% of cases and APS manifestations in 3.9%, 8.6%, and 6.9%, respectively [5]. The mechanism is mainly related to molecular mimicry [19]. If persistent or transient LA and/or aβ2GPI seem to have a similar impact on thrombosis development, in some studies, only the persistent aCL was associated with increased thrombotic risk [20]. Therefore, even if aPLs could rise during infections, a second measurement is needed to validate their persistence. The number of studies that tested aPL persistence in COVID-19 is small and showed constant positive levels, especially for IgM aCL and not for LA [21]. In this regard, similar to other infections, the rate of aPL persistence in COVID-19 seems to be low, less than 30% [22].

Regarding vaccination, one study reported conversion to aPL positivity at one month after the first dose of the HBV recombinant vaccine in 10% of cases [23]. Further, the human papillomavirus vaccine and typhoid one were involved in aPL positivity following administration [24].

Similarly, the aberrant activation of coagulation was rarely reported as a serious side effect following the COVID-19 vaccination [25]. The ACE2 receptor, the docking structure, is found in many tissues and might explain unwanted rare immune reactivations against the self, including the clotting system [26]. Depending on the type of vaccine used, different pathogenic mechanisms were proposed to explain aPL production. An adenoviral vector-based vaccine might induce aPLs due to molecular mimicry, the mRNA-based vaccine might trigger TLR activation or NETosis with the direct initiation of the coagulation cascade, and it might also disrupt, due to the liposomal structure, the interactions between the platelets and endothelial cells. The occurrence of anti-idiotype antibodies reacting against anti-spike antibodies might explain the coagulation disorders at 1–3 months after vaccination [25].

In our country, the vaccination was introduced in the last days of the year 2020, practically in the first three months of 2021 for medical staff and for the persons at high risk for severe COVID-19, and this coincided with the end of our study (all our included patients were unvaccinated).

The pro-thrombotic status of COVID-19 patients is well known and associated with disease prognosis [14]. Even if the lipid-binding aPL produced during SARS-CoV-2 infection has thrombotic and inflammatory effects [27], then the presence of aPL might be just an epiphenomenon and not the main mechanism for thrombosis development [14]. Of relevance, COVID-19-related coagulopathy is closely related to infection onset as well as to the associated inflammatory status. Additionally, if considering the APS pathology, thrombosis occurs later, after a longer time interval from the appearance of the APS pathologic antibodies [21]. Thus, the mechanism of coagulopathy and thrombosis occurrence in COVID-19 is complex and multifactorial, involving endothelial damage, inflammatory cytokines, hyperviscosity, possible thrombophilia mutations, and decreased platelet functions due to moderate thrombocytopenia as well as platelet surface dysfunction with a reduction in contractility [28].

Notably, aPLs are active in vitro anticoagulants, and prolonged clotting tests may mask the real hypercoagulopathy state [28]. In clinical practice, fibrinogen and D-dimer levels are the most informative indicators for increased coagulopathy [28]. Additionally, coagulation disorders in COVID-19 are not similar to those of sepsis and disseminated intravascular coagulation, which is primarily characterized by microthrombosis development [28]. Platelets are important players not only in hemostasis but also in inflammation and immune processes [29]. Thrombocytopenia occurs in viral infections secondary to immune-mediated pathways, intravascular disseminated coagulation, or micro/thrombi development [29,30]. Up to one-fifth of COVID-19 patients present thrombocytopenia, and the platelet number is associated with disease prognosis and amelioration with subsequent recovery [2].

Even if the aPL relation to overt thrombosis in COVID-19 still needs to be proven, there are data showing that aPL appearance is related to disease severity and inflammation degree in COVID-19 [17]. aPL was significantly associated with the CRP, ferritin [17,27], and D-dimer levels and was inversely related to lymphocyte count [27]. Additionally, COVID-19 patients who developed aPL had a longer duration of hospital stays and a higher need for oxygen therapy [17], but no differences were found regarding the mechanical ventilation need, intensive care admission rate, thrombosis, or mortality [17].

A potential limitation of this study is its observational nature. However, the parameters of interest in the present research were part of the standard care of COVID-19 patients in our department. Another limitation might be related to the laboratory methods used to test the aPLs, as there are constant discussions over the need for the standardization of APS tests, and the last updated guidelines included a mixing test for LA detection [31]. Even so, the kits used in our laboratory of immunology were among the assays tested and recommended for the APS assessment [32,33]. We do not present data about the persistence of aPLs for more than 12 weeks. Our research included patients without clinical APS criteria, and the COVID-19 population of interest is not usually tested for aPLs.

The association of aPLs with laboratory parameters of inflammation and coagulation as well as with COVID-19 disease severity advances a pathogenic role of aPLs in SARS-CoV-2 infection progression [27]. However, the available data are still inconclusive for considering these antibodies as part of COVID-19 management [15]. Even if testing aPLs in COVID-19 patients outside research purposes is not recommended, the data obtained could help to better understand both APS and COVID-19 mechanisms [34].

## 5. Conclusions

In summary, our study provides evidence of the occurrence of aPLs in SARS-CoV-2 infection in patients with no overt thrombotic event and of their correlation with COVID-19 severity.

## Figures and Tables

**Table 1 biomedicines-11-01241-t001:** Descriptive statistics of the patients included in the study.

Parameter	*n* = 179
Age, years mean ± SD	59.6 ± 14.5
Gender, F/M (%F)	99/80 (55.3)
Weight, kg mean ± SD	82.7 ± 17.9
Height, cm mean ± SD	168.3 ± 9.8
Disease severity	
Mild, *n* (%)	45 (25.1%)
Moderate, *n* (%)	62 (34.6%)
Severe, *n* (%)	72 (40.2%)
Pulmonary involvement (CT), % med (q1;q3)	30.0 (15.0; 40.0)
Pulmonary involvement (CT)	
None, *n* (%)	9/135 (5.0%)
Mild < 25%, *n* (%)	40/135 (22.3%)
Moderate 25–50%, *n* (%)	59/135 (33.0%)
Severe 50–75%, *n* (%)	21/135 (11.7%)
Very severe > 75%, *n* (%)	6 (3.4%)
Saturation O2, % mean ± SD	90.4 ± 10.3
Minimal saturation O2, % mean ± SD	86.6 ± 9.2
Necessary O2, l/min med (q1;q3)	3.0 (1.0; 7.0)
Period from the symptoms start, days med (q1;q3)	8.0 (5.0; 10.0)
Period from the RT-PCR SARS-CoV2 positive test, days med (q1;q3)	3.0 (1.0; 6.0)

COVID—severe acute respiratory syndrome coronavirus 2 disease; CT—computer tomography; F—female; M—male; SARS-CoV2—severe acute respiratory syndrome coronavirus 2; SD—standard deviation; RT-PCR—real time—polymerase chain reaction.

**Table 2 biomedicines-11-01241-t002:** Bivariate analysis in COVID-19 patients: lupus anticoagulant (LA)-positive versus LA-negative patients.

Parameter	LA+	LA−	*p*-Value
Age, years mean ± SD	60.1 ± 14.3	53.1 ± 16.2	0.125
Pulmonary involvement (CT), % med (q1;q3)	30.0 (20.0; 40.0)	20.0 (20.0; 20.0)	0.125
Anticardiolipin antibodies IgM, MPL med (q1; q3)	12.8 (10.1; 23.4)	7.77 (5.58; 12.69)	0.004
Anticardiolipin antibodies IgG, GPL med (q1; q3)	5.38 (3.65; 9.40)	6.6 (4.3; 11.4)	0.451
Anti-beta2 glycoprotein 2 I beta IgG, GBU med (q1; q3)	4.6 (2.6; 6.9)	2.6 (1.9; 4.0)	0.027
Ferritin, ng/mL med (q1;q3)	504 (240; 815)	143 (89; 436)	0.012
Lymphocytes/μL, med (q1;q3)	960 (710; 1380)	1870 (940; 2410)	0.040
CRP, mg/dL med (q1;q3)	44.2 (10.8; 85.8)	13.1 (5.7; 19.9)	0.027
Interleukin-6, pg/mL med (q1;q3) n = 84	27.6 (10.0; 59.5)	18.4 (21.3; 19.9)	0.622
ALAT, U/L med (q1;q3) n = 175	30.2 (19.4; 47.4)	32.8 (17.8; 49.5)	0.905
Albumin, g/dL med (q1;q3) n = 141	3.8 ± 0.5	3.9 ± 0.9	0.513
Creatin-kinase, U/L med (q1;q3) n = 175	100 (57; 185)	50 (36; 169)	0.179
Troponin, pg/mL med (q1;q3) n = 45	11.7 (7.4; 27.6)	7.0 (6.0; 8.0)	0.317
NT pro-BNP, pg/mL med (q1;q3) n = 95	335 (141; 1004)	294 (66; 635)	0.648
Creatinine, mg/dL med (q1;q3)	0.84 (0.69; 1.11)	0.75 (0.63; 0.89)	0.223
25OH-vitamineD, ng/mL med (q1;q3) n = 143	25.8 ± 13.1	31.1 ± 7.2	0.283

aCL—anticardiolipin antibodies; aB2GPI—anti-beta2 glycoprotein I beta; ALAT—alanine aminotransferase; CRP—C-reactive protein; GBU—microgram of IgG anti-beta2 glycoprotein I beta antibody per litre; GPL—microgram of IgG anticardiolipin antibody per litre; NT pro-BNP—N-terminal pro b-type natriuretic peptide; LA—lupus anticoagulant; p-value by Mann-Whitney test; MPL—microgram of IgM anticardiolipin antibody per litre.

## Data Availability

All available data was added in the Appendix A.

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
