# Peer review of "Antiphospholipid Antibodies Occurrence in Acute SARS-CoV-2 Infection without Overt Thrombosis"

_biomedicines, 2023, doi:10.3390/biomedicines11051241_

Round 1
Reviewer 1 Report
In this manuscript, the authors assess the presence of antiphospholipid antibodies (aPL), including lupus anticoagulant (LA), in COVID-19. They found a high level of LA in their patient cohort. One major limitation is the authors did not assess aPL persistence, and so their finding basically aligns to most studies performed in this space. I have the following additional specific comments/suggestions:
1. Abstract: line 24: “intensely positive” is an unusual description; would “strongly positive” be better?
2. Lines 58-59: “…understood, and aPL involvement in the thrombotic phenomena related to SARS-CoV-2 infection remains a matter of debate” the authors cite reference 9 here, which is a 2018 reference, so unlikely to be related to COVID-19. Also, on line 60: “the data remain scarce”. I would recommend they replace the citation for ref 9; for example, they could cite PMID: 34130340 and 34130341, which are extensive reviews on this topic) – also, the data related to aPL and COVID-19 is not really that scarce, and most studies did not show an association of aPL with thrombosis (as clearly identified in the above reviews [PMID: 34130340 and 34130341]).
3. Lines 70-71: “Furthermore, patients with a history of thrombophilia or autoimmune diseases or chronic anticoagulant treatment and patients with intra-COVID-19 thrombotic events were excluded from the analysis.” It is unclear if patients on acute anticoagulant treatment were also excluded. Most patients with COVID-19 are given anticoagulant treatment, and this may affect LA testing. Were all anticoagulant treated patients excluded? If not, how was the possibility of anticoagulant therapy managed within the study?
4. Lines 91-92: “20 MLP for aCL IgM, 20 GLP for aCL IgG, and 20 USG for aβ2GPI IgG.” I think the authors mean MPL and GPL for MLP and GLP respectively? What is ‘USG’? Authors should define the units used, not just provide the abbreviations.
5. Lines 93-98: “LA was detected by the standardized three-step procedure based on the guidelines of the International Society on Thrombosis and Hemostasis14 using the STA R Max® hemostasis analyzer developed by Diagnostica Stago. The LA integrated tests include screening and confirmation in a single procedure. The results were calculated by the LA ratio (screen/ confirm). The laboratory cut-offs used to define LA positivity were as follows: LA < 1.2 negative, LA 1.2 - 1.5 weakly positive, LA 1.5 - 2.0 positive, and LA > 2.0 intensely positive.” First, why did the authors not follow the latest guidelines on LA detection (2020; PMID: 33462974). The process may essentially be the same, but the authors should follow the latest guidelines. Also, the authors need to provide complete details on the reagents used; were these based on dRVVT or APTT, or preferably both?
6. Lines 192-195: “The number of studies that tested aPLs persistence in COVID-19 is small and showed constant positive levels, especially for IgM aCL and not for LA 23. In this regard, similar to other infections, the rate of aPLs persistence in COVID-19 seems to be low, less than 30%.” The 2 sentences seem contradictory. The first seems to imply persistence for IgM aCL and not LA, whereas the second suggests low level of persistence. See also: PMID: 34130340 and 34130341; these extensive reviews of the literature essentially indicate most studies did not assess persistence, and when assessed, most aPL were transient.
Minor:
1. Ref 2 citation should probably be updated; first published 2020, so publication should be finalized by now.
2. Supplementary data: ‘vitamineD’ should read ‘vitamin D’; ‘Intensely positive’ for LA – suggest authors use ‘strongly positive’ instead; supp Table 4: aPTT does not appear to be raised – this is good in so far as it seems to exclude anticoagulant therapy in the patient cohort, but ‘bad’ in so far as LA diagnosis is concerned, unless authors are using an LA resistant aPTT reagent? But then in Table 8 authors report correlation between aPTT and LA – again, authors need to clarify that they excluded anticoagulant effects in their patient cohort, or the correlation could simply be of false LA due to anticoagulant therapy? Table 9: platelets not plateles.
Author Response
Dear Editor and Reviewers,
On behalf of my co-authors, I thank you very much for the time dedicated to the manuscript “Antiphospholipid antibodies occurrence in acute SARS-CoV-2 infection without overt thrombosis”.
We would like to express our gratitude for the assessment of our manuscript, the valuable suggestions made in your review notes and for the opportunity of a revision.
Therefore, we have revised the document and made the corrections suggested by the reviewers.
Kindly find below a point-by-point response to your comments.
REVIEWER 1
COMMENT 1
In this manuscript, the authors assess the presence of antiphospholipid antibodies (aPL), including lupus anticoagulant (LA), in COVID-19. They found a high level of LA in their patient cohort. One major limitation is the authors did not assess aPL persistence, and so their finding basically aligns to most studies performed in this space. I have the following additional specific comments/suggestions:
- Abstract: line 24: “intensely positive” is an unusual description; would “strongly positive” be better?
Response 1
Thank you for your comment. We have replaced “intensely positive” with “strongly positive” throughout the manuscript.
COMMENT 2
- Lines 58-59: “…understood, and aPL involvement in the thrombotic phenomena related to SARS-CoV-2 infection remains a matter of debate” the authors cite reference 9 here, which is a 2018 reference, so unlikely to be related to COVID-19.
Response 2
Thank you very much for your observation, we cited by mistake another article by the same first author. We further reviewed all references according to reviewers’ comments.
COMMENT 3
Also, on line 60: “the data remain scarce”. I would recommend they replace the citation for ref 9; for example, they could cite PMID: 34130340 and 34130341, which are extensive reviews on this topic) – also, the data related to aPL and COVID-19 is not really that scarce, and most studies did not show an association of aPL with thrombosis (as clearly identified in the above reviews [PMID: 34130340 and 34130341]).
Response 3
We are grateful for help in improving our manuscript. We changed the entire paragraph and added the two references proposed.
COMMENT 4
- Lines 70-71: “Furthermore, patients with a history of thrombophilia or autoimmune diseases or chronic anticoagulant treatment and patients with intra-COVID-19 thrombotic events were excluded from the analysis.” It is unclear if patients on acute anticoagulant treatment were also excluded. Most patients with COVID-19 are given anticoagulant treatment, and this may affect LA testing. Were all anticoagulant treated patients excluded? If not, how was the possibility of anticoagulant therapy managed within the study?
Response 4
Thank you for this observation. Preventive anticoagulation was added to treatment in almost all COVID-19 patients hospitalized in our department (e.g. exception when thrombopenia, risk of hemorrhage etc.). However, the APS serology was part of the first blood sample at admission, before any treatment starts. One supplementary phrase was added to the manuscript to clarify this aspect.
There were sometimes 6 or 8 admissions at the same time, patients brought at the same time from the emergency rooms of other hospitals. Therefore, the protocol in our department was standardized, data regarding the anticoagulation received were precisely noted in all files to avoid, for example given anticoagulation once more. Even if the study is done retrospectively, the available data registered prospectively during the patients’ admission were extended collected (including for example the check list for any possible COVID-related symptoms) as available in the Supplementary Files.
COMMENT 5
- Lines 91-92: “20 MLP for aCL IgM, 20 GLP for aCL IgG, and 20 USG for aβ2GPI IgG.” I think the authors mean MPL and GPL for MLP and GLP respectively? What is ‘USG’? Authors should define the units used, not just provide the abbreviations.
Response 5
Thank you very much for this observation, we are sorry that we missed translating the units’ abbreviations into English. We replaced the manuscript MLP with MPL and GLP with GPL. We also provided the abbreviations in Table 2 (one MPL unit refers to 1 microgram of IgM anticardiolipin antibody, while one GPL unit refers to 1 microgram of IgG anticardiolipin antibody). Further USG was replaced with GBU (abbreviation noted also in Table 2: one GBU unit refers to 1 microgram of IgG anti-beta2 glycoprotein 2 I beta antibody).
COMMENT 6
- Lines 93-98: “LA was detected by the standardized three-step procedure based on the guidelines of the International Society on Thrombosis and Hemostasis14 using the STA R Max® hemostasis analyzer developed by Diagnostica Stago. The LA integrated tests include screening and confirmation in a single procedure. The results were calculated by the LA ratio (screen/ confirm). The laboratory cut-offs used to define LA positivity were as follows: LA < 1.2 negative, LA 1.2 - 1.5 weakly positive, LA 1.5 - 2.0 positive, and LA > 2.0 intensely positive.” First, why did the authors not follow the latest guidelines on LA detection (2020; PMID: 33462974). The process may essentially be the same, but the authors should follow the latest guidelines. Also, the authors need to provide complete details on the reagents used; were these based on dRVVT or APTT, or preferably both?
Response 6
Thank you for your comments. The period for inclusion in this study and APS antibodies testing was April 2020 – May 2021. The new guideline you proposed was published after the study starts, in November 2020. However, we used validated methods for the APS tests, the same reagent used in other studies reporting results on aPLs (e.g. PMID: 31487100; PMID: 29623883; PMID: 35312119). The reagents used for the LA detections were based on both dRVVT and APTT, and it was noted in the methods.
COMMENT 7
- Lines 192-195: “The number of studies that tested aPLs persistence in COVID-19 is small and showed constant positive levels, especially for IgM aCL and not for LA 23. In this regard, similar to other infections, the rate of aPLs persistence in COVID-19 seems to be low, less than 30%.” The 2 sentences seem contradictory. The first seems to imply persistence for IgM aCL and not LA, whereas the second suggests low level of persistence. See also: PMID: 34130340 and 34130341; these extensive reviews of the literature essentially indicate most studies did not assess persistence, and when assessed, most aPL were transient.
Response 7
Thank you for your comment. Indeed, we were not clear here – what we wanted to say is that among the studies that tested aPLs, just a few (30%) showed aPLs persistence, and the majority (70%) showed transient aPLs. We changed the paragraph in the manuscript in order to clarify this idea.
COMMENT 8
Minor
- Ref 2 citation should probably be updated; first published 2020, so publication should be finalized by now.
Response 8
Thank you for your comment, we completed reference 2.
COMMENT 9
- Supplementary data: ‘vitamineD’ should read ‘vitamin D’; ‘Intensely positive’ for LA – suggest authors use ‘strongly positive’ instead; supp Table 4: aPTT does not appear to be raised – this is good in so far as it seems to exclude anticoagulant therapy in the patient cohort, but ‘bad’ in so far as LA diagnosis is concerned, unless authors are using an LA resistant aPTT reagent? But then in Table 8 authors report correlation between aPTT and LA – again, authors need to clarify that they excluded anticoagulant effects in their patient cohort, or the correlation could simply be of false LA due to anticoagulant therapy? Table 9: platelets not plateles.
Response 9
We thank you for this observation, the spelling mistakes have been corrected in the supplementary files.
Regarding the correlations, we used LA resistant aPTT reagent and we excluded anticoagulation therapy. I think it is to note that the degree of correlations is low, with a rho coefficient less than 0.300 (a relation that might be due to the background inflammatory state). There were taken all measures to avoid a method/ laboratory error.
Thank you for your positive and constructive feedback regarding the topic we address in our review.
We remain open to any further corrections to the manuscript.
Best regards,
The authors

Reviewer 2 Report
The article ""Presence of antiphospholipid antibodies in acute SARS-CoV-2 infection without overt thrombosis" reports observations on patients with SARSCoV-2 infection, focusing on the role of aPL, known to have already been described as increased in these patients, without, however, suggesting a role for this increase.
The work is interesting in some respects (Comparison of LA positivity or negativity and other laboratory parameters) while it is less interesting when trying to shed light on the possible relationship between aPL and degree of severity of COVID, a finding that is always conflicting in different studies.
The work deserves a review on the discussion of methodological (assay method and CRP levels) and clinical aspects, discussing the role of aPLs in the infection phase, after recovery and following vaccination.
Author Response
Dear Editor and Reviewers,
On behalf of my co-authors, I thank you very much for the time dedicated to the manuscript “Antiphospholipid antibodies occurrence in acute SARS-CoV-2 infection without overt thrombosis”.
We would like to express our gratitude for the assessment of our manuscript, the valuable suggestions made in your review notes and for the opportunity of a revision.
Therefore, we have revised the document and made the corrections suggested by the reviewers.
Kindly find below a point-by-point response to your comments.
REVIEWER 2
COMMENT
The article "Presence of antiphospholipid antibodies in acute SARS-CoV-2 infection without overt thrombosis" reports observations on patients with SARSCoV-2 infection, focusing on the role of aPL, known to have already been described as increased in these patients, without, however, suggesting a role for this increase.
The work is interesting in some respects (Comparison of LA positivity or negativity and other laboratory parameters) while it is less interesting when trying to shed light on the possible relationship between aPL and degree of severity of COVID, a finding that is always conflicting in different studies.
The work deserves a review on the discussion of methodological (assay method and CRP levels) and clinical aspects, discussing the role of aPLs in the infection phase, after recovery and following vaccination.
Response 1
Thank you for reviewing our manuscript and for the valuable comments. We think that the advantage of this study over others is that there are presented data from the pandemic beginning, (we started in April 2020). We had a very rigorous protocol at admission (data presented in the supplementary files) - with clinical, laboratory, and imagistic data, very rigorous registered from the admission, thus our patients were well characterized compared to other data from the same period.
We also thank you for the suggestion to further extend the discussion over follow-up and vaccinations. As result, new paragraphs were added in the discussion section. In our country, the vaccination was introduced in the last days of the year 2020, practically in the first three months of 2021 for medical staff and for the persons at high risk for severe COVID-19 (and not for the whole population), and this coincided with the end of our study - all our included patients were unvaccinated. We added this information in the manuscript text.
Thank you for your positive and constructive feedback regarding the topic we address in our review.
We remain open to any further corrections to the manuscript.
Best regards,
The authors

Round 2
Reviewer 1 Report
The authors have addressed my major concerns; some minor issues remain:
1. The authors have revised the terminology for the aPL (lines 99-101; Table 2 footer); however, for MPL and GPL these values are per litre (or /L); hence the MPL (IgM per litre) and GPL (IgG per litre) abbreviations
2. Authors need to identify the reagents used for LA testing. Saying: "LA was detected by the standardized three-step procedure based on the guidelines of the International Society on Thrombosis and Hemostasis10 using the STA R Max® hemostasis analyzer developed by Diagnostica Stago." is not enough. What reagents were used? Were both dRVVT and APTT assays used for LA testing? Please clarify.
3. Line 134: "None of the patients included had the vaccine COVID-19". Suggest, "None of the patients included had been vaccinated against COVID-19".
Author Response
Dear Editor and Reviewers,
We thank you for giving us an opportunity to revise our manuscript titled “Antiphospholipid antibodies occurrence in acute SARS-CoV-2 infection without overt thrombosis” (Manuscript ID: diagnostics- 2307925).
We have followed the comments and revised the manuscript accordingly.
REVIEWER 1
The authors have addressed my major concerns; some minor issues remain:
COMMENT 1
- The authors have revised the terminology for the aPL (lines 99-101; Table 2 footer); however, for MPL and GPL these values are per litre (or /L); hence the MPL (IgM per litre) and GPL (IgG per litre) abbreviations .
Response 1
Thank you for your comment. We did the changes noted for the abbreviations.
COMMENT 2
- Authors need to identify the reagents used for LA testing. Saying: "LA was detected by the standardized three-step procedure based on the guidelines of the International Society on Thrombosis and Hemostasis10 using the STA R Max® hemostasis analyzer developed by Diagnostica Stago." is not enough. What reagents were used? Were both dRVVT and APTT assays used for LA testing? Please clarify.
Response 2
Thank you for these comments. We asked our laboratory and the reagents used are now noted in the methods. We also added a new phrase for the study’s limitations and a new reference was added.
COMMENT 3
- Line 134: "None of the patients included had the vaccine COVID-19". Suggest, "None of the patients included had been vaccinated against COVID-19".
Response 3
Thank you for this observation. The phrase was changed as proposed.
We remain open to any further corrections to the manuscript.
Best regards,
The authors
